# The Use of Iron(II,III) Oxide (Fe_3_O_4_) as a Cross-Linking Agent for Unfilled and Filled Chlorosulfonated Polyethylene (CSM) and Study of the Vulcanizates Properties

**DOI:** 10.3390/ma15207276

**Published:** 2022-10-18

**Authors:** Aleksandra Smejda-Krzewicka, Konrad Mrozowski, Piotr Kobędza, Agnieszka Adamus-Włodarczyk

**Affiliations:** 1Institute of Polymer and Dye Technology, Faculty of Chemistry, Lodz University of Technology, Stefanowskiego Street 16, 90-537 Lodz, Poland; 2Institute of Applied Radiation Chemistry, Faculty of Chemistry, Lodz University of Technology, Wroblewskiego Street 15, 93-590 Lodz, Poland

**Keywords:** chlorosulfonated polyethylene, iron(II,III) oxide, cross-linking, filler, morphology, Payne effect, flammability, oxygen index

## Abstract

This paper discusses the cross-linking behaviors, mechanical and dynamical properties, and flammability of elastomeric composites containing unconventionally cured chlorosulfonated polyethylene (CSM). The purpose of this work was to verify the CSM ability to cross-link with iron(II,III) oxide (Fe_3_O_4_) and to produce flame retardant materials. During the first series of tests, three types of CSM were used, differing in the content of bound chlorine (29–43%). The results showed that the CSM with 43% bound chlorine (Hypalon 30, CSM43) was the most advantageous type of chlorosulfonated polyethylene in terms of its properties. It exhibited a short vulcanization time, a high degree of cross-linking, and very good mechanical properties. In the next stage, the CSM composites with various fillers (talc, arsil, kaolin, chalcedonite, or carbon black) were prepared, because filled rubber materials are of the greatest practical importance. The cross-linking kinetics, equilibrium swelling, mechanical and dynamic properties as well as flammability were studied. It was found that the addition of fillers led to a decrease in the degree of cross-linking, an increase in the vulcanization time (in the case of talc, arsil, or kaolin), an increase in the overall mechanical strength (in the case of carbon black, arsil or talc). All filled vulcanizates proved to be non-flammable, as the specific oxygen index value exceeded 37.5%.

## 1. Introduction

Chlorosulfonated polyethylene (CSM) is a special elastomer containing chlorosulfonic and chlorinated groups. This polymer contains 25–45% bound chlorine and 1.0–2.2% bound sulfur [1,2,3,4,5]. To ensure the best processing properties of CSM, the chlorine content in the macromolecules should not exceed 27%, while the chlorosulfonic groups should not exceed 1.5% [4]. The chlorosulfonated polyethylene structure consists of ethylene-, chloroethylene-, and chlorosulfonated ethylene units (Figure 1) [4,5]. 

CSM can be obtained by the modification and functionalization of polyethylene (PE) [6]. It consists in the simultaneous action of chlorine and sulfur dioxide (dissolved in the CCl_4_ solution), in the presence of radical initiators, on PE [6,7]. The technological and functional properties as well as the chemical reactivity and cohesion characteristics of CSM depend on the content of bound chlorine and sulfur. Lower-density CSM dissolves well in aromatic and chlorinated hydrocarbons [1,2,3,4,5,8,9,10,11,12,13,14,15,16]. It is noteworthy that the substitution of some number of hydrogen atoms in polyethylene chain with chlorine atoms and chlorosulfonic groups causes a disruption of the regularity of the polymer chain structure, and as a result the crystallization susceptibility changes [1]. As a result, the flexibility of CSM macromolecules is increased. Based on the literature reports, as well as on structural considerations, it can be stated that bound chlorine and chlorosulfonic groups have a favorable effect on the thermal properties of the elastomer and flame retardancy of its vulcanizates [10,11,12,13,14,16]. However, the main problem is the thermal decomposition of chlorosulfonated polyethylene. The products of combustion pose a risk during a fire because they are harmful to human health and life and for ecological reasons [4,17]. The contents of chlorosulfonic and chlorine groups also provide high resistance to ozone, oxidative aging, and thermo-oxidation. CSM is also characterized by high hardness and significant tensile strength [10,11].

For cross-linking, oxides of metals of constant valence (zinc, magnesium [18]) are mainly used. Non-standard methods with oxides of metals of various valence can also be used, however, one should be careful because they influence on accelerated aging of vulcanizates [18,19,20]. According to the literature, metal hydroxides, stearates, abiethates, naphthenates, aliphatic diamines, and thiocarbamates are also used for vulcanization [1,2,3]. The cross-linking with metal oxides leads to the formation of an ionic spatial network and consequently to good mechanical and functional properties of the cross-linked products [19,20,21]. Moreover, the presence of chemically active groups in the CSM macromolecules makes it possible to use it as a component of self-cross-linking elastomeric compositions. Such compositions include blends of CSM with carboxylated acrylonitrile–butadiene rubber (CSM/XNBR) or acrylonitrile–butadiene rubber (CSM/NBR) [8,9,10,11,12].

Due to its very good thermal properties and high-temperature resistance caused by bound chlorine and its good miscibility with other rubbers, chlorosulfonated polyethylene is used as the outer layer of conveyor belts for transporting hot materials [1,4]. CSM is also used to manufacture heat-resistant gaskets, spacers for presses, and foamed articles [22,23]. However, the greatest use of rubber made from CSM has been in the automotive industry. It is used as a lining for cables and wires due to its good thermal and resistance to organic solvents, oils, and fuels. In the building industry, CSM is used to manufacture roofing membranes, liners, and covers for portable water tanks [22,23,24,25].

One of the most important requirements for rubber products is the resistance to higher temperatures of 300–500 °C. As a result of technological developments, there is an increased need for rubbers that can withstand the environment, where the temperature is high and very often causes irreversible changes in the mechanical properties of rubber [1,2,3]. Therefore, the purpose of this study was to obtain flame-retardant materials based on chlorosulfonated polyethylene and non-standard cross-linking of CSM with iron(II,III) oxide. The iron(II,III) oxide was used as a new unconventional cross-linking agent of chlorosulfonated polyethylene. This is due to the need to find an alternative to ZnO. European Union regulations force a significant limitation of the use of ZnO in various fields, including elastomer technology. The regulations result from the harmful effects of ZnO on aquatic organisms. Alternatives to ZnO include various metal oxides that can be used as cross-linking agents [18,26]. In this study, three different types of CSM were used to determine which one had the best vulcametric and mechanical properties. Various quantities of bound chlorine affect the performance of chlorosulfonated polyethylene, so CSM with 29% bound chlorine, 35% and 43% had to be studied. Then, the chosen type of chlorosulfonated polyethylene (CSM43) was filled with five various fillers and the mechanical, dynamic and vulcametric properties, the equilibrium swelling, morphology, Payne effect, and the flammability of the produced vulcanizates were studied. The fillers used are talc, arsil, kaolin, chalcedonite, and carbon black. These compounds have been chosen due to the fact that four of them are minerals, which theoretically should reduce the flammability of the compounds and improve mechanical properties. However, the literature reports that one of the fillers (carbon black) increases the thermal and electrical conductivity of the elastomer and does not reduce the flammability of the vulcanizate [26,27,28,29]. In addition to the tests done in the first series, flammability and dynamic properties were tested. The dynamic parameters of the vulcanizates were determined to find out how the fillers dispersed in the elastomer matrix, while the flammability test using the oxygen index method was intended to check whether the stated aim of the work had been achieved.

## 2. Materials and Methods

### 2.1. Materials

In this study, different types of chlorosulfonated polyethylene (CSM) were used:Hypalon 20 (CSM29), product of DuPont de Nemours Inc. (Wilmington, Delaware, USA) with bound chlorine content of 29% by weight and a density of 1.12 g/cm^3^,Hypalon 30 (CSM43), product of DuPont de Nemours Inc. (Wilmington, Delaware, USA) with bound chlorine content of 43% by weight and a density of 1.26 g/cm^3^,Hypalon 40 (CSM35), product of DuPont de Nemours Inc. (Wilmington, Delaware, USA) with bound chlorine content of 35% by weight and a density of 1.18 g/cm^3^.

Iron(II,III) oxide, Fe_3_O_4_ (Sigma-Aldrich Sp. z o. o., Poznan, Poland) with a particle size < 5µm, pureness > 95%, density of 5.2 g/cm^3^, and specific surface (BET) of 6.56 m^2^/g was used as a cross-linking agent. Stearic acid, SA (Chemical Worldwide Business Sp. z o. o., Slupca, Poland) with a density of 0.85 g/cm^3^ was used as a dispersing agent.

The following materials were used as fillers:precipitated silica Arsil (Z. Ch. Rudniki S.A., Rudniki, Poland), with a bulk density ~0.15 g/cm^3^ and pureness > 95%,technical kaolin (POCH S.A., Gliwice, Poland), with a density of 2.60 g/cm^3^ and an average grain size of 1.3 µm,carbon black, CB (Chemical Worldwide Business Sp. z o. o., Slupca, Poland), with a bulk density of 180 g/cm^3^,chalcedonite (CRUSIL Sp. z o. o., Inowlodz, Poland), with a density of 2.56 g/cm^3^ and a particle size < 10 µm,talc (KOCH Co., Ltd., Seoul, Korea), with a density of 2.78 g/cm^3^.

The following solvents were also used: diethyl ether (Chempur, Piekary Slaskie, Poland, product, with a density of 0.71 g/cm^3^), toluene (POCh S.A., Gliwice, Poland, product, with a density of 0.87 g/cm^3^), heptane (POCh S.A., Gliwice, Poland, product, with a density of 0.68 g/cm^3^).

### 2.2. Methods

CSM composites were prepared using a Krupp-Gruson laboratory two-roll mill with a roll diameter of 200 mm and a length of 450 mm. The temperature of the roll was 20–25 °C, while the speed of the front roll was 20 rpm, with the rolls friction of 1:1.25. The study was divided into two series. In the first series of tests, 3 unfilled blends with different elastomer matrices (different types of CSM) were prepared. For the first series of tests the rubbers were plasticized, then the stearic acid and the iron(II,III) oxide were incorporated. The preparation of one blend lasted 5 min. The prepared 3 mixtures were stored separately in a hermetically closed foil at room temperature and then it was conditioned for 24 h. The second series of tests were performed in the same way as the first; however, 5 filled compositions containing HYPALON 30, stearic acid, Fe_3_O_4_, and fillers (kaolin, talc, chalcedonite, arsil, or carbon black) were prepared. Each filler was used in an amount of 30 phr. The process of formation looked as: first, stearic acid with iron(II,III) oxide was added, and then the fillers. The total blending time was longer than in the first series, as it ranged from 5 to 10 min. The obtained samples were also stored in a hermetically closed foil, each separately.

Vulcametric measurements were determined by the Alpha Technologies MDR 2000 rotorless rheometer, heated to 160 °C. The oscillation frequency was 1.67 Hz. The test lasted 60 min and was performed according to ASTM D5289-17 standard [30]. The test consisted of registering the torque (*M*) as a function of time (*t*) during the cross-linking of the test sample at a constant temperature (*T*). The torque value depends on the stiffness of the rubber mixture and changes as the cross-linking process progress. Based on the vulcametric curves *M* = f (*t*), the optimal cross-linking time (*t*_90_—the time at which the torque reaches 90% of the increase), the scorch time (*t*_02_), the minimum torque (*M*_min_), and the torque increment after a specified heating time (Δ*M_t_*) were determined. The cure rate index (*CRI*), a measure of the cross-linking rate, was calculated from Formula (1) and the torque increment after a given time of heating was calculated from Formula (2):(1)$$CRI=\frac{100}{t_{90}-t_{02}}$$
(2)$$\Delta M_{x}=M_{x}-M_{min}$$

The prepared samples were then vulcanized using an electrically heated hydraulic press. The vulcanization parameters for the first series of tests were as follows: temperature—160 °C, pressure—200 bars, time—15 min. For the second series of measurements, the vulcanization parameters were as follows: temperature—160 °C, pressure—200 bars, time—5 min for arsil, kaolin, chalcedonite, and carbon black, 15 min for talc. The preparation of the samples was based on the cutting of a molded mix of approximately 20 g. For easier removal of molds, poly(tetrafluoroethylene) (PTFE, Teflon) foil was removed after the process was used.

The determination of equilibrium swelling was performed. Samples were cut from prepared vulcanizates in four different shapes. Each of them weighed 25 to 50 mg, with an accuracy of 0.1 mg. The samples were then placed with solvents: toluene or heptane, in weighing vessels. The prepared samples were placed in a thermostatic chamber for 72 h at 25 °C, which was then bathed with diethyl ether, dried on filter paper, and then weighed again. The samples were then dried in a dryer at a temperature of 50 °C to a constant weight and they were reweighed. The equilibrium volume swelling in toluene or heptane (*Q_v_*) was calculated from Formula (3):(3)$$Q_{v}=Q_{w}\cdot \;\frac{d_{v}}{d_{s}}\;$$
where: *Q_w_* is the value of equilibrium weight swelling (mg/mg); *d*_v_ is the vulcanizate density (g/cm^3^), *d*_s_ is the solvent density (g/cm^3^). 

The equilibrium weight swelling was calculated from Formula (4):(4)$$-Q_{w}=\frac{m_{s}-m_{d}}{m_{d}^{*}}$$
where: *m*_s_ is the swollen sample weight (mg); *m*_d_ is the dry sample weight (mg); $m_{d}^{*}$ is the reduced sample weight calculated from Formula (5):(5)$$m_{d}^{*}=m_{d}-m_{0}\cdot \frac{m_{m}}{m_{t}}$$
where: *m*_0_ is the initial sample weight (mg); *m*_m_ is the mineral substances content in the compound (mg); *m*_t_ is the total weight of the compound (mg).

The rubber volume fraction (*V_r_*) was calculated from Formula (6):(6)$$V_{r}=\frac{1}{1+Q_{V}}$$

The degree of cross-linking (α_c_) was determined from Formula (7):(7)$$\alpha_{c}=\frac{1}{Q_{v}}$$
where: *Q_v_* is the equilibrium volume swelling (ml/mL).

The tensile properties were measured according to the PN-ISO 37: 2017 [31] standard using a ZwickRoell machine (model 1435, Ulm, Germany) connected with the appropriate computer software. In the study, samples in the shape of B-type paddles with a measuring section width of 4 mm were used. The scope of the properties tests included: stress at 100%, 200%, 300% of elongation (*S*_e100_, *S*_e200_, *S*_e300_), tensile strength (*TS*_b_), and elongation at break (*E*_b_).

The tear strength (T_s_) was measured using the A method in accordance with the PN-ISO 34-1: 2010 standard using a ZwickRoell machine (model 1435, Ulm, Germany) connected with appropriate computer software. Rectangular-shaped samples with the following dimensions 100 mm × 15 mm were used for the tests.

The hardness (HA) was measured with a ZwickRoell (Ulm, Germany) hardness tester in accordance with ISO 48-4: 2018. The samples for this test were prepared in the shape of cylinders in a specially prepared form. The measurement results were determined on the Shore A scale.

Measurements of the Payne effect (Δ*G*′), maximum storage modulus (*G*′_max_), and maximum loss modulus (*G"*_max_) were made on vulcanizate discs with a diameter of 25 mm and a thickness of 2 mm using the TA Instruments ARES G2 rotational rheometer (Newcastle, UK) according to ISO 4664: 2011. The value of the Payne effect (ΔG′) was calculated according to Formula (8):(8)$$\Delta G^{\prime }=\Delta G\prime_{\max}-\Delta G\prime_{\min}$$
where: *G*′_max_ is a maximum value of the storage modulus [MPa], *G*′_min_ is a minimum value of the storage modulus [MPa].

The surface morphology of the fillers and filled vulcanizates was evaluated using a scanning electron microscope (SEM) Hitachi Tabletop Microscope TM-1000 (Tokyo, Japan). The preparation of samples for measurement consisted of placing a double-sided self-adhesive foil on special tables and gluing the tested sample to it. Then, a gold layer was applied to the prepared sample using the Cressington Sputter coater 108 auto vacuum sputtering machine (Redding, CA, USA) at a pressure greater than 40 mbar, for 60 s. The sample prepared in this way was placed in a scanning electron microscope chamber and the measurement was performed. Using the TM-1000 software, the results were recorded on a computer that cooperated with the spectrophotometer.

The flammability of vulcanizates was determined by the oxygen index (*OI*) method. Vulcanizates samples of 50 × 10 × 4 mm were placed vertically in the holder and covered with a quartz column. The construction of the apparatus is protected by the Polish Patent [32]. Inside the column, the sample was perfused with a composition of oxygen (O_2_) and nitrogen (N_2_). The gas flow rate was determined by rotameters. Nitrogen flow was constant and reached 400 L/h, while the oxygen flow was variable and selected to determine the lowest oxygen concentration in the gas composition at which the sample burned during 180 ± 15 s. Samples were ignited for 5 s with a gas burner. After the removal of the fire source, the time of their combustion was measured. The test was performed according to PN-ISO 4589-2. The oxygen index (*OI*) was calculated from Formula (9):(9)$$OI=\frac{O_{2}}{O_{2}+N_{2}}\;\cdot 100\%$$
where: O_2_ is the oxygen flow rate [L/h] and N_2_ is the nitrogen flow rate [L/h].

The time of burning in air (*t*_b_) was determined using the same samples as for the oxygen index. The vertically positioned samples were ignited for 5 s using a gas burner. After this time, the following parameters were measured: the burning time of the sample or the time after which the sample was extinguished. The measurement was repeated five times.

## 3. Results and Discussion

### 3.1. Influence of Bound Chlorine Content on CSM Cross-Linking with Iron(II,III) Oxide and Properties of Its Vulcanizates

To investigate the ability of chlorosulfonated polyethylene to cross-link with iron(II,III) oxide, 3 mixes containing various types of CSM, differing in the content of bound chlorine were prepared. Iron(II,III) oxide (3 phr of Fe_3_O_4_) and stearic acid (1 phr of SA) were added to the chlorosulfonated polyethylene. Table 1 shows the compositions of these mixes.

An important aspect to select the right type of CSM is the vulcametric parameters, which determine the viscosity of the composite and determine the scorch time (t_02_), the vulcanization time (t_90_), and the degree of cross-linking (α_c_). Therefore, the length of the vulcanization time is known and the most optimal one is chosen. Table 2 shows the vulcametric parameters of the tested elastomeric mixes. 

The results in Table 2 clearly show that the quantity of bound chlorine influences the vulcametric properties. The compound with the highest amount of bound chlorine (CSM43) has the shortest vulcanization time (t_90_ = 16.05 min), meaning it cross-links the fastest. This fact may indicate the participation of bound chlorine in effective cross-linking of CSM with Fe_3_O_4_. The more chlorine bound in the CSM macromolecule, the more efficient the vulcanization process of this elastomer is. Moreover, it has the highest degree of cross-linking, which can be seen by the highest values of torque increment after 10, 15, or 20 min of heating (12.12 dNm, 13.49 dNm, 14.55 dNm). The minimum torque indicating the viscosity of the mixes is at a similar level (M_min_ ~0.31 dNm) among the three mixes, so it is possible to conclude that the quantity of bound chlorine does not significantly affect the viscosity. The other two composites (CSM29 and CSM35) show a lower torque value and have a much longer vulcanization and scorch time (t_90_ = 37.32 min and t_90_ = 35.32 min). In the case of the scorch time, this is advantageous due to safe processing, whereas in terms of the vulcanization time, this is not optimal because the longer the time, the longer the vulcanization conditions have to be maintained, which proves costly in industrial production. The CSM43 mix obtains the highest value of vulcanization rate index (CRI = 6.53 min^−1^). The vulcametric curve of the CSM43 shows the highest torque moment after 25–30 min, after which the torque moment is significantly reduced. This phenomenon indicates the reversion, i.e., the breakdown of cross-linkages forming during the cross-linking. This means a reduction in the degree of cross-linking if the curing time exceeded 30 min. Most likely it is related to the largest bound chlorine content (43% by weight) for the CSM43 (Figure 2). 

An equilibrium swelling study of the CSM vulcanizates tested shows that the amount of bound chlorine influences the degree of cross-linking of the final product. The CSM43 vulcanizate (with the highest content of bound chlorine) has the highest degree of cross-linking, as indicated by the Q_v_ (1.15 mL/mL) and V_r_ (0.465) values (Table 3). The value of Q_v_ in both heptane and toluene is lower than this parameter calculated for the other two vulcanizates. Furthermore, it can be established that chlorosulfonated polyethylene is more sensitive to the interaction through toluene than through heptane, as each of the tested vulcanizates shows higher values for Q_v_ and for -Q_w_ (for CSM29: Q_v_^T^ = 2.71 mL/mL, Q_v_^H^ = 0.62 mL/mL).

Tensile strength is one of the main mechanical properties describing rubber materials. From a physical point of view, it is the stress corresponding to the highest tensile force obtained during a stable tensile test. Tensile testing is performed according to PN-ISO 37:2017 on a universal testing machine Zwick-Roell model 1435 (Ulm, Germany). The tensile strength results describe the stiffness, the mechanical strength, and the elongation at break. These are important parameters defining rubber materials, as they determine their usefulness and application in given fields. 

The results of the test performed are presented in Table 4. It was shown that the CSM43 vulcanizate achieves the highest values of tensile strength at break (TS_b_ = 6.97 MPa), and the other CSM vulcanizates have lower TS_b_ values oscillating between 1.2 and 1.7 MPa. Only one of the samples (CSM29) reaches a relative elongation above 100%, so it can be concluded that the tested vulcanizates are extremely stiff. Summarizing all the observations, we have found that CSM43 vulcanizate, despite its very high stiffness, is characterized by good mechanical strength, while the other vulcanizates have neither quite high elasticity nor high mechanical strength. Perhaps this is due to the morphology of the rubber products tested.

The scanning electron microscopy (SEM) analysis reveals the surface morphology of the tested CSM vulcanizates (Figure 3). The image of the CSM29 vulcanizate shows large ripples and unevenness (Figure 3a). Figure 3b shows the uniform elastomer phase, in which small clusters of iron oxide(II,III) are visible. Conversely, large flat structures (lamellae) are unevenly distributed in the CSM35, which indicates that the components are not dispersed properly in the elastomer matrix (Figure 3c). The SEM examination results and analysis of properties of obtained products indicate that the CSM43 is the most appropriate elastomer for further research.

### 3.2. Cross-Linking Characteristics and Properties of Filled and Cross-Linked CSM with 43% Bound Chlorine

After the first series of tests, in which we checked the cross-linking ability of different chlorosulfonated polyethylene with iron(II,III) oxide with the best possible performance properties, different fillers to produce flame retardant products were used. We have chosen CSM43 and it was filled with various fillers, with the purpose of creating flame retardant products. The fillers used were arsil, kaolin, chalcedonite, talc, or carbon black. The compositions of the mixes prepared for further analysis are shown below (Table 5).

Before incorporating the fillers into the CSM43 mixes, their morphology was examined from SEM (Figure 4). The silica particles are fine aggregated and agglomerated (Figure 4a). Kaolin has much larger flat grains, which also form larger clusters. In many places, the kaolin particles are arranged parallel to each other (Figure 4b). The chalcedonite particles are completely different than the grains of the other fillers used. They have oblong, needle-like shapes and are loosely coupled (Figure 4c). Talc has large and flat grains (even >30 µm), and there are many empty areas between them (Figure 4d). Carbon black is characterized by single particles in the form of lamellae. There are very large agglomerates and a lot of void space where the elastomer chains can penetrate (Figure 4e).

After studying the cross-linking kinetics and processing the results accordingly (Table 6), it was observed that the selection of a suitable filler could improve the degree of cross-linking and vulcanization properties to different extents, but could also increase the cure time and affect the scorch time.

Each filled sample obtains a higher viscosity and a longer scorch time than the unfilled sample. Among the compositions tested, the compound filled with arsil shows the highest minimum torque (M_min_ = 1.19 dNm) and the highest degree of cross-linking in terms of torque increments after 5 and 10 min of heating (21.46 dNm, 26.19 dNm, respectively). However, the vulcanization time for the mixes containing arsil or talc is the longest (t_90_~47 min) of those tested. This is due to the vulcametric curve and its marching modulus. At the increasing torque, the curing time determined with the rheometer is calculated in proportion to the test duration. The CSM43 filled with kaolin shows a much shorter vulcanization time (t_90_ = 30.94 min) than the CSM43-A or CSM-T. The addition of chalcedonite or carbon black to chlorosulfonated polyethylene results in a significant reduction in the vulcanization time (3.45 min, 5.17 min, respectively) at the expense of a lower degree of cross-linking compared to other fillers. Such large differences in the kinetics curves of CSM43 prove the large share of the tested fillers in the process of cross-linking of this elastomer. Considering only the cross-linking kinetic, carbon black or chalcedonite are the most preferred fillers for CSM43.

Calculated from Equation (1) (Section 2.2), the cure rate index (CRI) shows that the fastest curing mixes are CSM43-Ch and CSM43-CB (CRI = 40.0 min^−1^, 29.58 min^−1^, respectively). The other filled chlorosulfonated polyethylene compositions have lower CRI values than unfilled mixes of CSM43. This indicates a longer processing time for this material. The vulcanization curve for CSM43-T shows a marching characteristic, while the other compositions tend to reverse. Due to the short vulcanization time, the increase in torque increment after 20 min of heating (ΔM_20_) determines the amount of reversion. CSM43-A and CSM43-K mixes (ΔM_20_ = 32.16 dNm, 23.72 dNm, respectively) show higher reversion during the longer heating than blends filled with chalcedonite and carbon black (12.53 dNm, 13.11 dNm, respectively). Figure 5 shows the curves for the filled CSM43 compositions. During the extended heating, the CSM43-K, CSM43-CB, and CSM43-A composites are reversed and started to degrade as indicated by the torque values.

The equilibrium swelling results obtained show that the addition of fillers decreases the degree of cross-linking of CSM43. The presence of mineral fillers such as arsil, kaolin, or chalcedonite increases the Q_v_ values relative to the unfilled vulcanizate in both toluene (1.57 mL/mL, 2.04 mL/mL, 2.12 mL/mL, respectively) and heptane (respectively: 0.13 mL/mL, 0.20 mL/mL 0.21 mL/mL), indicating a reduction in the degree of cross-linking. Moreover, the eluted substance values, i.e., -Q_w_, increase with respect to the unfilled vulcanizate, this is due to the degree of cross-linking as mentioned earlier, because the more cross-linked the vulcanizate, the less sensitive to solvents. Table 7 shows the results from the equilibrium swelling study of filled CSM43 vulcanizates.

The CSM43-A, CSM43-T, and CSM43-CB vulcanizates show the highest degree of cross-linking (α_c_~0.65) among all the filled vulcanizates. Summarizing the results, it is easy to see that despite the different fillers the values of the following parameters: –Q_w_ and V_R_ and Q_v_ have similar values. However, relative to the unfilled vulcanizate these values are different (for unfilled CSM: Q_v_^T^ = 1.15 mL/mL–Q_w_^T^ = 0.03, V_R_^T^ = 0.465, while for talc-filled CSM43: Q_v_^T^ = 1.47 mL/mL, –Q_w_^T^ = 0.26, V_R_^T^ = 0.405). The advantage in the α_c_ value of the unfilled composition can be observed easily.

The results of the tensile strength test (Table 8) show that the filler can affect both the stiffness and flexibility of the vulcanizate. The talc-filled sample increases its overall mechanical strength by almost double but remained the stiffest among the samples (with extended vulcanization time). Vulcanizates filled with chalcedonite or kaolin show an increase in stiffness, but a decrease in tensile strength. The CSM43-A vulcanizate has increased stiffness and mechanical strength (S_e100_ = 7.05 MPa, TS_b_ = 12.40 MPa). The vulcanizate with arsil has the E_b_ value of 187%, which compares to the unfilled vulcanizate, this parameter is improved by more than 4 times, and TS_b_ increases by almost 2 times. Carbon black as a filler causes a slight increase in mechanical strength (TS_b_ = 7.82 MPa). CSM43-Ch and CSM43-K vulcanizates have decreased mechanical strength relative to the unfilled compound. Summarizing the results, it can be clearly stated that the addition of arsil has the most positive effect on the tensile strength of the vulcanizate, as its presence in the CSM43 contributes to the increased mechanical strength of the product. This fact may be due to the very large fragmentation of arsil (compare with Figure 4a), large specific surface area, and its strong interactions with CSM43.

Another parameter determined for rubber materials is tearing – a parameter that determines the durability of these products. The test is based on the fact that two forces applied in an antagonistic direction cause the tearing of the material. Tear strength is defined as the maximum force acting on a material to tear it. Among the rubbers tested, talc-filled vulcanizate (CSM43-T) shows the highest tear resistance (Table 8) (7.77 N/mm). Very similar results were obtained for samples with chalcedonite, kaolin, and arsil (6.56 N/mm, 6.46 N/mm, and 6.10 N/mm, respectively). The worst-performing sample is the CSM43 filled with carbon black (CSM43-CB, T_s_ = 4.35 N/mm).

The hardness of rubber materials is measured on the Shore A scale. This term refers to a static method for measuring the hardness of rubber, elastomers, and rigid plastics. It was invented by the aforementioned Albert F. Shore, who developed an instrument, called a durometer. During the present study, a type A durometer was used and the results obtained can be found in Table 8. In elastomeric materials, the factor that influences the hardness is the degree of cross-linking of the elastomer. The higher the α_c_, the higher the hardness of the vulcanizate. The vulcanizate characterized by the highest hardness is the CSM43-T product (HA = 92.5 °ShA). The highest hardness of the sample containing talc is most likely due to the prolonged cross-linking time and large agglomerates of talc distributed in the CSM matrix. The other samples reach a high level of hardness for cross-linked elastomers, as the values of this parameter are in the range of 75–85 °ShA (Table 8). It is worth noting that all filled CSM43 vulcanizates had above-average hardness, which largely depends on the hardness and stiffness of pure elastomer. This is the cause of the difficulties observed in the processing of chlorosulfonated polyethylene.

The dynamic properties of rubber materials depend on the type of rubber, and the amount and activity of the filler used. They result directly from the strength of the rubber–filler and filler–filler interactions, the so-called “extra network” is formed, which has a great influence on the dynamic properties. Storage modulus (G′) is a measure of the elasticity of the rubber, while the loss modulus (G″) determines the ability to dissipate energy and convert it into heat. 

The results presented in Table 9 suggest that the talc-filled vulcanizate has large concentrations of filler particles. The maximum values of the storage modulus (G′_max_) and loss (G″_max_) for CSM43-T are 13.1 MPa and 10.2 MPa, respectively, while for the other samples the G′_max_ and G″_max_ values are in the range of 0.96–1.41 MPa and 0.55–1.09 MPa, respectively. These values probably indicate the formation of agglomerates in the CSM43-T vulcanizate, due to the large value of G″_max_. The maximum values of G′ and G″ coincide with the plateau, therefore they can be considered the same. This observation is also confirmed by the SEM image of the vulcanizate with talc (Figure 4d), in which large and flat agglomerates of this filler were clearly visible.

The curves for chalcedonite, kaolin, carbon black, and arsil are very similar on the graph showing the dependence of the storage modulus on oscillation (Figure 6). The CSM43-Ch sample takes the largest G′ values among them. The curve of CSM43-T is different from the others—there is a sharp decrease in G′ values on the graph; therefore, ΔG for talc is a much higher value than for other samples.

A similar trend for the loss modulus is shown in Figure 7. Analogically, the vulcanizate filled with talc shows much higher values than the other fillers and also a visible significant decrease in the G″ value with increasing oscillation. For the other four vulcanizates, the curves are very similar to the graph showing the storage modulus (Figure 6). The Payne effect for CSM43-K, CSM43-A, CSM43-S, and CSM43-Ch samples is much less than for talc-filled chlorosulfonated polyethylene.

Figure 8 shows SEM images of surfaces of filled CSM43 vulcanizates. Figure 8a presents the morphology of the CSM43 filled with arsil. Most likely, the correct dispersion of the silica in the elastomer matrix results from the large fragmentation of this filler and its small grains, which is shown in Figure 4a. In the CSM43 filled with kaolin (Figure 8b), single, large clusters of this filler are visible, which were also shown in Figure 4b. Figure 8c shows the folded structure of CSM43 vulcanizate containing chalcedonite. There are large aggregates and agglomerates of chalcedonite, which is most likely due to its specific shape (Figure 4c). Figure 8d presents large, improperly dispersed talc agglomerates in the elastomer in some areas, most likely caused by large talc particles (Figure 4d). In the center of the SEM image of the CSM43 filled with carbon black, a large agglomerate of this filler (>10 µm) is visible (Figure 8e). The tendency of carbon black to form large clusters is also confirmed in Figure 4e.

Determination of the flammability of rubber by means of the oxygen index method is very common but also does not require advanced apparatus to measure the flammability of polymer materials. The advantages of this method are the repeatability of results and the high accuracy. The oxygen index (OI) is important for rubber materials because it provides information on the flammability of the material. Flammable materials achieve an OI of less than 21%, i.e., they burn in air. Flame retardant materials among polymers should be considered those with an OI ≥ 28%. The flammability test results show unequivocally that CSM43 vulcanizates made with any filler are non-flammable (Table 10).

Chlorosulfonated polyethylene is one of the materials with limited combustibility (OI = 27%); however, the addition of mineral fillers gave the vulcanizates a non-flammable character. The incorporation of silica and kaolin allows the formation of a filler–rubber network, which contributed to improved flame retardancy. Even the carbon black-filled vulcanizate achieves properties similar to the rest, despite the fact that carbon black is not a flame retardant but actually supports combustion. Each sample achieves OI ≥ 37.5%, as this is the maximum value that could be obtained on this apparatus (the limit was reached in terms of oxygen flow in the combustion chamber). The exact oxygen index values of each of these compositions are not known; however, it can be concluded that they are certainly flame retardants. Furthermore, each of the tested samples extinguishes as soon as the fire was removed, indicating that the filled CSM43 vulcanizates proposed in this paper are self-extinguishing materials. In addition to the addition of suitable fillers, the amount of bound chlorine in the chlorosulfonated polyethylene has a positive effect on the flame retardancy of the material.

In the case of tested fillers, the mechanism of increasing the flame resistance is based on three functions. Firstly, the incorporation of a filler results in a dilution of the polymer. This means that for a certain volume of filled material there is less fuel to keep the fire going than for the same volume of unfilled material. Secondly, when materials containing kaolin and talcum are burned, ash is formed around the filler particles into scales, which reduces further fire propagation. Thirdly, mineral fillers contain bound water in their structure, which additionally increases the fire resistance of the entire material. These three functions of the mineral fillers combine to create a synergistic effect, further increasing the material’s flame-retardant efficiency.

## 4. Conclusions

In the first series of tests, it was shown that the CSM43 (chlorosulfonated polyethylene with the highest chlorine-bound content) is the type of CSM which has the best physical and mechanical properties. Testing the cross-linking progress on a rheometer and the results obtained indicates that the CSM43 mix is cured the fastest (t_90_ = 16.05 min, CRI = 6.53 min^−1^) and it has the highest degree of cross-linking confirmed by the highest torque increment (ΔM_10_ = 12.12 dNm). Furthermore, the results of the strength properties clearly show that the vulcanizate made from HYPALON 30 has the best mechanical strength (TS_b_ = 6.97 MPa), while an equilibrium swelling test showed that CSM with 43% of bound chlorine is the best cross-linked rubber (Q_v_^T^ = 1.15 mL/mL, Q_v_^H^ = 0.10 mL/mL, α_c_ = 0.87). Moreover, it is possible to unconventionally cross-link the composites studied with iron(II,III) oxide. As a result, this substrate is found to be useful for the cross-linking of chlorosulfonated polyethylene.

The chosen proper type of CSM43 was used to produce flame-retardant materials. The results of the cross-linking kinetics of filled composites show that the filler, depending on the type, can have different effects on the vulcanization time. Chlorosulfonated polyethylene filled with chlacedonite or carbon black cross-links shorter (3.45 min and 5.17 min, respectively) than unfilled CSM43 (16.05 min). The addition of talc, kaolin, or arsil causes increases in vulcanization time, which is an undesirable effect in the processing industry context. Results of equilibrium swelling of CSM43 vulcanizates clearly show that the unfilled vulcanizate (–Q_w_^T^ = 0.03 mg/mg, –Q_w_^H^ = 0.10 mg/mg) is more resistant to toluene and heptane than filled (–Q_w_^T^ = 0.28 mg/mg, –Q_w_^H^ = 0.19 mg/mg). The compositions with talc (α_c_ = 0.68), arsil (α_c_ = 0.64), or carbon black (α_c_ = 0.62) have the highest values of the degree of cross-linking among the filled samples, however, the addition of filler causes a decrease in the degree of cross-linking with respect to the unfilled CSM43 (α_c_ = 0.87). Strength parameters and dynamic properties were also studied including the Payne effect. Vulcanizates filled with talc and arsil show high values of TS_b_ parameter (11.00 MPa and 12.40 MPa, respectively). The CSM43 vulcanizate with talc is much harder than the other samples, while the composition with arsil is more flexible, but its strength properties are at a similar level to the vulcanizate with talc. Due to the addition of kaolin or chalcedonite to the chlorosulfonated polyethylene, the tensile strength values decrease (4.76 MPa, 4.81 MPa, respectively). The talc-filled CSM43 is the hardest (92.5°ShA) and has the highest tear strength (T_s_ = 7.77 N/mm).

Investigations of the dynamic property tests show that kaolin, arsil, carbon black, and chalcedonite are properly dispersed in the elastomer matrix. Only the talc-filled vulcanizate has Payne effect values much higher than the other fillers (for CSM43 with talc: ΔG′ = 13.1 MPa). The SEM analysis shows that talc agglomerates in the elastomer matrix are formed.

The filling of chlorosulfonated polyethylene with adequate substances causes flame retardation of the CSM43 vulcanizate. All the compositions tested show an increase in the oxygen index to a minimum of 37.5%. This value confirms that these materials are non-flammable even in an environment where the oxygen concentration would be around 37%. In addition, when the fire source is removed from the sample, the flame is immediately extinguished, proving the self-extinguishing nature of the filled products. In conclusion, the objective of our work has been achieved, as each of the filled CSM43 vulcanizates is flame retardant and it was possible to cross-link this elastomer in a non-standard way using iron(II,III) oxide.

## Figures and Tables

**Figure 1 materials-15-07276-f001:**
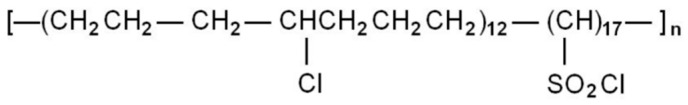
Formula of chlorosulfonated polyethylene.

**Figure 2 materials-15-07276-f002:**
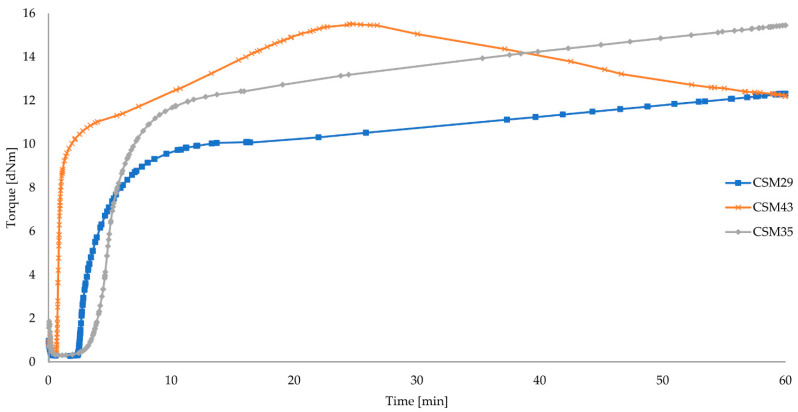
Vulcametric kinetics of chlorosulfonated polyethylene cross-linked with iron(II,III) oxide (3 phr of Fe_3_O_4_).

**Figure 3 materials-15-07276-f003:**
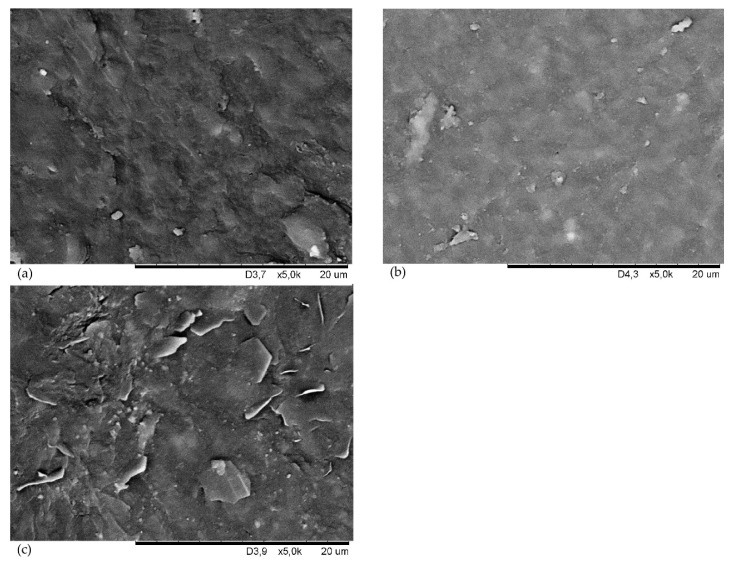
Scanning electron microscope (SEM) images of the vulcanizates containing different types of chlorosulfonate polyethylene: CSM29 (**a**), CSM43 (**b**), CSM35 (**c**).

**Figure 4 materials-15-07276-f004:**
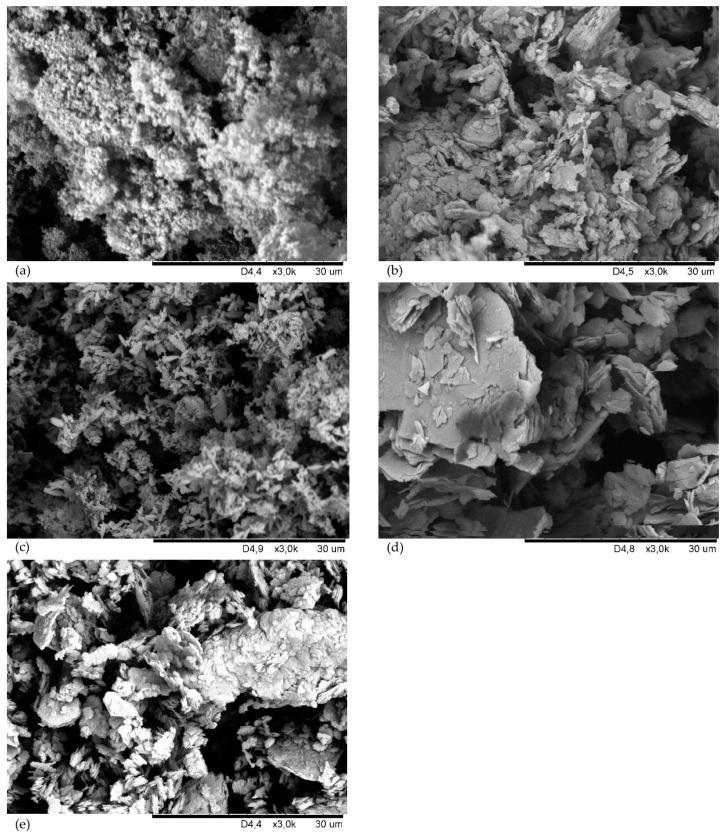
Scanning electron microscope (SEM) images of the fillers used into CSM43: arsil (**a**), kaolin (**b**), chalcedonite (**c**), talc (**d**), carbon clack (**e**).

**Figure 5 materials-15-07276-f005:**
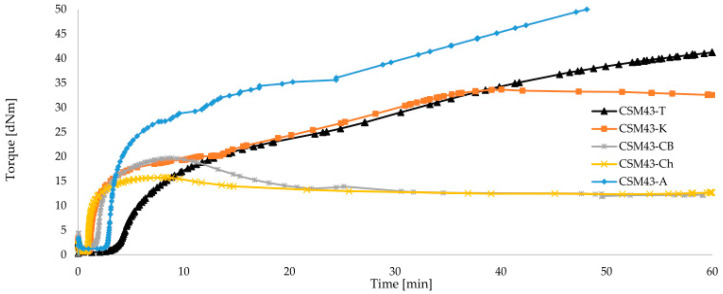
Vulcametric kinetics of filled chlorosulfonated polyethylene cross-linked with iron(II,III) oxide (3 phr of Fe_3_O_4_).

**Figure 6 materials-15-07276-f006:**
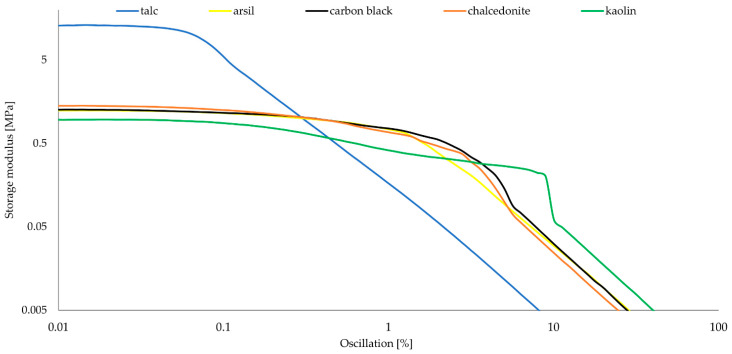
Storage modulus of filled CSM43; T = 160 °C.

**Figure 7 materials-15-07276-f007:**
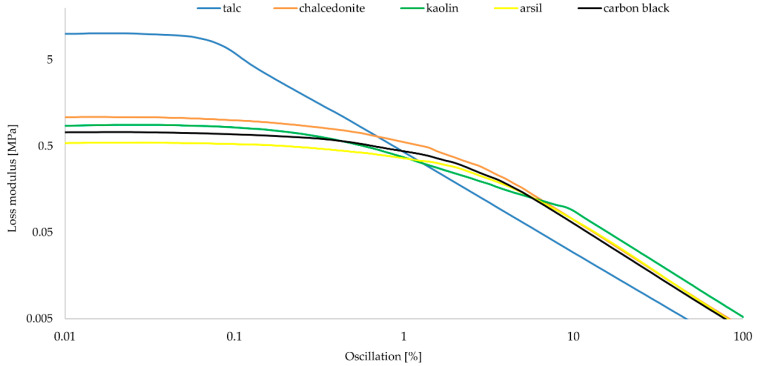
Loss modulus of filled CSM43; T = 160 °C.

**Figure 8 materials-15-07276-f008:**
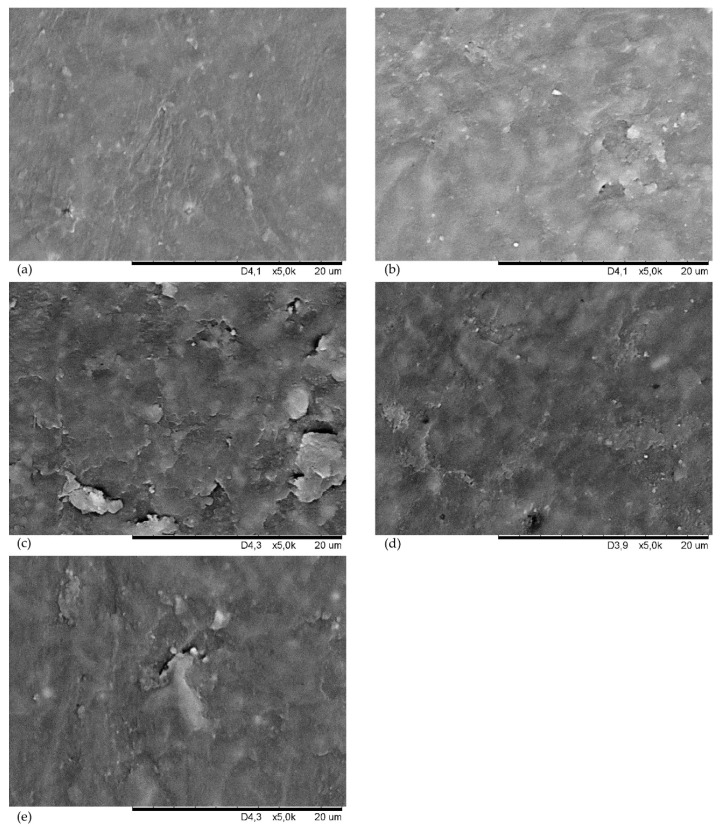
Surface morphology SEM images of CSM43 vulcanizates filled with: arsil (**a**), kaolin (**b**), chalcedonite (**c**), talc (**d**), or carbon black (**e**).

**Table 1 materials-15-07276-t001:** The formulations of the composites used in the first series of tests.

Component Content (phr)			
Hypalon 20	100	---	---
Hypalon 30	---	100	---
Hypalon 40	---	---	100
Fe_3_O_4_	3	3	3
SA	1	1	1
Symbol	CSM29	CSM43	CSM35

**Table 2 materials-15-07276-t002:** Vulcametric parameters of the unfilled CSM compositions determined at the temperature of 160 °C.

Symbol	M_min_	ΔM_10_	ΔM_15_	ΔM_20_	t_02_	t_90_	CRI	t_v_	Vulcametric Curves
	dNm	dNm	dNm	dNm	min	min	min^−1^	min	
CSM29	0.27	9.28	9.82	10.04	2.72	37.32	2.89	15	marching
CSM43	0.36	12.12	13.49	14.55	0.74	16.05	6.53	15	reversion
CSM35	0.30	11.37	12.12	12.42	4.12	35.32	3.20	15	marching

*M_min_*—minimum torque, Δ*M_10_*, Δ*M_15_*, Δ*M_20_*—torque increment after 10, 15 or 20 min of heating, *t_02_*—scorch time, *t_90_*—cure time, *CRI*—cure rate index, t_v_—vulcanization time.

**Table 3 materials-15-07276-t003:** Values of equilibrium swelling of unfilled CSM vulcanizates; T = 160 °C, t = 15 min.

Symbol	Toluene				Heptane		
	Q_v_^T^ (cm^3^/cm^3^)	–Q_w_^T^ (mg/mg)	V_R_^T^	α_c_ (-)	Q_v_^H^ (cm^3^/cm^3^)	–Q_w_^H^ (mg/mg)	V_r_^H^
CSM29	2.71 ± 0.04	0.09 ± 0.01	0.269 ± 0.003	0.37	0.62 ± 0.03	0.07 ± 0.01	0.616 ± 0.011
CSM43	1.15 ± 0.03	0.03 ± 0.01	0.465 ± 0.006	0.87	0.10 ± 0.02	0.04 ± 0.01	0.911 ± 0.009
CSM35	2.56 ± 0.05	0.07 ± 0.01	0.281 ± 0.004	0.39	0.34 ± 0.02	0.05 ± 0.01	0.747 ± 0.008

*Q_v_^T^*, *Q_v_^H^*—equilibrium volume swelling in toluene or heptane, –*Q_w_^T^*, –*Q_w_^H^*—equilibrium weight swelling in toluene or heptane, *V_R_^T^*, *V_R_^H^*—volume fraction of rubber in swollen material, *α_c_*—degree of cross-linking, determined for toluene.

**Table 4 materials-15-07276-t004:** Mechanical properties of unfilled CSM vulcanizates; T = 160 °C, t = 15 min.

Symbol	*S_e100_* [MPa]	*TS_b_* [MPa]	*E_b_* [%]
CSM29	1.55 ± 0.05	1.61 ± 0.10	106 ± 9
CSM43	---	6.97 ± 0.56	38 ± 12
CSM35	---	1.25 ± 0.10	88 ± 4

*S_e100_*—stress at elongation 100%, *TS_b_*—tensile strength, *E_b_*—elongation at break.

**Table 5 materials-15-07276-t005:** The formulations of the composites used in the second series of tests.

Component Content (phr)						
Hypalon 30 (CSM43)	100	100	100	100	100	100
Fe_3_O_4_	3	3	3	3	3	3
SA	1	1	1	1	1	1
Arsil	30	---	---	---	---	---
Kaolin	---	30	---	---	---	---
Chalcedonite	---	---	30	---	---	---
Talc	---	---	---	30	---	---
Carbon black	---	---	---	---	30	---
Symbol	CSM43-A	CSM43-K	CSM43-Ch	CSM43-T	CSM43-CB	CSM43

**Table 6 materials-15-07276-t006:** Vulcametric parameters of the filled CSM43 compositions determined at the temperature of 160 °C.

Symbol	*M_min_*	Δ*M_5_*	Δ*M_10_*	Δ*M_20_*	*t_02_*	*t_90_*	Vulcametric Curves	*CRI*	t_v_
	dNm	dNm	dNm	dNm	min	min	---	min^−1^	min
CSM43-A	1.19	21.46	26.19	32.16	2.86	47.12	marching	2.26	5
CSM43-K	0.63	16.75	14.68	23.72	1.16	30.94	reversion	3.36	5
CSM43-Ch	0.68	14.45	14.77	12.56	0.95	3.45	reversion	40.0	5
CSM43-T	0.51	7.30	16.46	22.45	3.97	46.52	marching	2.35	15
CSM43-CB	0.97	16.55	18.43	13.11	1.79	5.17	reversion	29.58	5
CSM43	0.36	12.12	13.49	14.55	0.74	16.05	reversion	6.53	15

Explanation of symbols: under Table 2.

**Table 7 materials-15-07276-t007:** Values of equilibrium swelling of filled CSM43 vulcanizates; T = 160 °C, t_v_ = 5–15 min.

Symbol	Toluene			Heptane			
	*Q_v_^T^* (cm^3^/cm^3^)	–*Q_w_^T^* (mg/mg)	*V_R_^T^*	*Q_v_^H^* (cm^3^/cm^3^)	–*Q_w_^H^* (mg/mg)	*V_r_^H^*	*α_c_*
CSM43-A	1.57 ± 0.04	0.28 ± 0.01	0.337 ± 0.009	0.13 ± 0.01	0.26 ± 0.01	0.883 ± 0.004	0.64
CSM43-K	2.04 ± 0.07	0.28 ± 0.01	0.329 ± 0.008	0.20 ± 0.04	0.26 ± 0.01	0.834 ± 0.024	0.49
CSM43-Ch	2.12 ± 0.05	0.28 ± 0.01	0.320 ± 0.005	0.21 ± 0.04	0.26 ± 0.01	0.83 ± 0.03	0.47
CSM43-T	1.47 ± 0.05	0.26 ± 0.01	0.405 ± 0.008	0.27 ± 0.02	0.26 ± 0.01	0.78 ± 0.01	0.68
CSM43-CB	1.62 ± 0.03	0.28 ± 0.01	0.381 ± 0.004	0.14 ± 0.03	0.25 ± 0.01	0.878 ± 0.023	0.62
CSM43	1.15 ± 0.03	0.03 ± 0.01	0.465 ± 0.006	0.10 ± 0.01	0.04 ± 0.01	0.911 ± 0.009	0.87

Explanation of symbols: under Table 3.

**Table 8 materials-15-07276-t008:** Mechanical properties of filled CSM43 vulcanizates; T = 160 °C, t_v_ = 5–15 min.

Symbol	*S_e100_* (MPa)	*TS_b_* (MPa)	*E_b_* (%)	*Ts* (N/mm)	*HA* (°ShA)
CSM43-A	7.05 ± 0.09	12.40 ± 0.70	187 ± 10	6.10 ± 0.41	82.8 ± 1.4
CSM43-K	5.90 ± 0.26	4.76 ± 0.38	82 ± 6	6.46 ± 0.78	80.7 ± 1.6
CSM43-Ch	3.83 ± 0.08	4.81 ± 0.28	140 ± 10	6.56 ± 0.74	77.1 ± 1.8
CSM43-T	---	11.00 ± 0.60	46 ± 9	7.77 ± 0.35	92.5 ± 0.8
CSM43-CB	---	7.82 ± 0.82	86 ± 9	4.35 ± 0.24	79.2 ± 1.4
CSM43	---	6.97 ± 0.56	46 ± 2	---	---

Explanation of symbols: under Table 4; *T_s_*—maximum tear strength, *HA*—hardness on Shore A scale.

**Table 9 materials-15-07276-t009:** Payne effect of filled CSM43 vulcanizates; T = 160 °C, t_v_ = 5–15 min.

Symbol	*G*′*_max_* (MPa)	*G*″*_max_* (MPa)	Δ*G* (MPa)
CSM43-A	1.24	0.55	1.24
CSM43-K	0.96	0.88	0.96
CSM43-Ch	1.41	1.09	1.41
CSM43-T	13.1	10.2	13.1
CSM43-CB	1.27	0.73	1.27

Δ*G*—Payne effect, *G*′*_max_*—maximum storage modulus, *G*″*_max_*—maximum loss modulus.

**Table 10 materials-15-07276-t010:** Flammability of filled CSM43 vulcanizates; T = 160 °C, t_v_ = 5–15 min.

Symbol	*t_b_* (s)	*OI* (%)	Observation
CSM43-A	<10	≥37.5	When the fire is applied, a “cold fire” effect occurs, when the fire is removed the flame goes out immediately.
CSM43-K	<10	≥37.5	The samples do not react to fire (they do not burn), the flame goes out immediately.
CSM43-Ch	<10	≥37.5	The samples fade when the fire is removed, a slight surface incandescence that subsides after a while.
CSM43-T	<10	≥37.5	The samples did not ignite, no glow, when the fire is removed the sample immediately goes out.
CSM43-CB	<10	≥37.5	The samples do not burn, after removal of the flame there is a slight glow which disappears after a while.

*t_b_*—time burning, *OI*—oxygen index.

## Data Availability

Data sharing not applicable.

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
