# Peer review of "The Use of Iron(II,III) Oxide (Fe3O4) as a Cross-Linking Agent for Unfilled and Filled Chlorosulfonated Polyethylene (CSM) and Study of the Vulcanizates Properties"

_materials, 2022, doi:10.3390/ma15207276_

Round 1
Reviewer 1 Report
-Not enough background information & problem statement was inclued in the abstract.
'was to obtain flame retardant materials based on chlorosul-fonated polyethylene and non-standard cross-linking of CSM with iron(II,III) oxide. Due to restrictions imposed by the European Union, a replacement for ZnO as a cross-linking agent is looked for.'
-Why was ZnO restricted by the European Union? Why was iron oxide and the other materials were chosen as the filler?
'The fillers used are talc, arsil, kaolin, chalcedonite and carbon black.'
'However, one of the fillers (carbon black) is literature...'-> 'is' should be 'in'
'As fillers of elastomer composites were used...' ->hanging statement
Methodology was performed according to various standards, and was described in detail.
Results include SEM images which describe the morphology of the CSM43 mixes.
DIscussion on effect of the filler on the stiffness and flexibility of the vulcanizate was also discussed in detail.
Overall, the manuscript can be accepted for publication after revisions to the abstract and elaboration to the justification for the use of the fillers.
Reviewer 2 Report
In this work, the authors study the ability of chlorosulfonated polyethylene (CSM) cross-linked with iron(II,III) oxide for flame retardant materials. Below are some questions and comments that need to be addressed.
1. What is the molecular weight of chlorosulfonated polyethylene and how would molecular weight affect flame retardant behavior?
2. In figure 2, CSM43 mix show highest peak at about 35 minutes of vulcametric time and continues to decrease. What is the rationale behind this?
3. Following the above question, CSM43 is mistakenly referred to as CSM 45, the authors need to double check this for consistency.
4. In Figure 3c, is the flake on the surface due to the structure or it is due to the sample preparation process, e.g. cutting the surfaces? The scale bar in the SEM images need to be labeled larger.
5. I suggest reporting the plateau value for G’ and G’’ instead of the maximum value. The G’ and G’’ curves show about the same characteristics, especially the high oscillation frequency, this is not very usual for viscoelastic materials. More comments are needed here.
6. Following the above comment, the author could also compute the G' and G" terminal decay slope as a function of oscillation frequencies and see if any rheological constitutive model can be used to explain the measured behavior.
7. For observations of fire retardant capabilities. Only descriptions by text doesn’t seem to be enough. The author should at least consider adding some pictures of comparison before, during, and after fire test for each sample.
Reviewer 3 Report
Journal: Materials
Manuscript ID: materials-1935313-peer-review-v1
Title: The use of iron(II,III) oxide (Fe3O4) as a cross-linking agent for
unfilled and filled chlorosulfonated polyethylene (CSM) and study of the vulcanizates properties
This manuscript studies the ability of chlorosulfonated polyethylene (CSM) cross-linked with iron(II,III) oxides to produce flame retardant materials by adding different fillers (talc, arsil, kaolin, chalcedonite or carbon black). I have carefully examined the manuscript. I recommend the manuscript for rejected in Materials. My comments are as follows.
Comments:
1. The introduction section is poorly logical. For example, “The reason for this phenomenon is the accelerated course of many destructive chemical processes (bond breakdown, rubber oxidation). CSM is a special elastomer containing chlorosulfonic and chlorinated groups.”. The authors should reorganize the logic of the introduction.
2. In introduction, what is the cross-linking method of iron(II,III) oxides in this manuscript, and what are its advantages and disadvantages compared to zinc oxide? There are many researches on the use of iron oxide as a cross-linking agent. The authors need to introduce this research progress.
3. In 2.2, what is the amount of each of the five fillers added?
4. In 3.1, what are reversal curves and rolling curves? What are the reasons for inversion curves or rolling curves?
5. In Results and Discussion, the authors only describe the phenomena and do not analyze the causes of these phenomena. For example, why is the vulcanization time shorter with higher bound chlorine content? How does bound chlorine affect the crosslinking of CMS rubber? What is the reason for the uneven dispersion of CMS35 elastomers? There are other similar questions which aren’t listed here. Please check carefully throughout paper.
6. In 3.2, “After studying the cross-linking kinetics and processing the results accordingly (Table 6), ...... time and affect the scorch time” and “The equilibrium swelling results obtained show that the addition of fillers decreases the degree of cross-linking of CSM”, why are the crosslinking kinetic results different from the swelling equilibrium results?
7. What is the flame retardant mechanism of each filler?
8. Basically, the authors should read the manuscript carefully. There are some format errors in the article. For example, in references 33, here is lack of page numbers; in 3.1, “The CSM45” should be “The CSM35”.

Round 2
Reviewer 2 Report
My questions and comments are addressed in the revised version.
Reviewer 3 Report
This manuscript studieds the ability of chlorosulfonated polyethylene (CSM) cross-linked with iron(II,III) oxides to produce flame retardant materials by adding different fillers (talc, arsil, kaolin, chalcedonic or carbon black) to increase their flame retardant properties.
I have carefully examined the revised manuscript. The authors have revised the manuscript according to my comments, and I recommend the manuscript for publication in Materials.